

# A machine learning approach to the geomorphometric detection of ribbed moraines in Norway

Thomas J. Barnes[1], Thomas V. Schuler[1], Simon Filhol[1], Karianne S. Lilleøren[1]

[1]Department of Geosciences, University of Oslo, 0316, Oslo, Norway

*Correspondence to*: Thomas J. Barnes (thomas.barnes@geo.uio.no)

**Abstract.**

Machine learning is a powerful yet underutilised tool in geomorphology, commonly used for image–based pattern recognition. Analysing new high–resolution (1 – 10 m) elevation datasets, we investigate its usefulness for detecting discrete geomorphological features. This study develops a machine learning–based method for identifying ribbed moraines in digital

elevation data and progresses to test its performance versus time consuming, manual methods. Ribbed moraines share geomorphometric characteristics with other glacial landforms, hence represent a valuable test of our new methodology in terms of differentiating between similar features, and wider for detection of landforms with similar characteristics. Furthermore, mapping ribbed moraines may provide valuable indications of their origin, a topic of debate within glacial geomorphology. To automatically detect ribbed moraines, we extract simple morphometrics from high–resolution digital elevation model data and

mask regions where ribbed moraines are unlikely to form. We then test several machine learning algorithms before examining the best performer (K–means clustering) on three study areas in Norway of 15 km$^2$. Our results demonstrate balanced accuracy of 65 – 75 % when validating versus ground–truth. The performance depends on the availability of high–resolution elevation data in Norway, needed to resolve the spatial scale of the target (10-100m). We find the method effective at detecting both fields of ribbed moraines as well as individual ribbed moraines. We propose pathways for future implementation of this method

on a large–scale and for increasing the detail of information gained about detected landforms. In conclusion, we demonstrate K–means clustering as a promising method for detecting ribbed moraines, with great potential to reduce the time needed to produce landform maps.






## 1 Introduction

### 1.1 Geomorphological mapping

Mapping of landforms has traditionally been a manual process, either through direct field observations or manual analysis of remotely sensed data (Smith & Clark, 2005; Verstappen, 2011; Evans, 2012; Sommerkorn, 2020). More recently,
semi–automated methods have been developed, where computational analysis of remote sensing data is interpreted by the operator (Guitet et al 2013), giving rise to subjectivity and human error (Saha et al., 2011; Eisank et al., 2014; Sommerkorn, 2020). Often, the restriction of data availability, quality and resolution have been the primary limitations leading to the maintenance of traditional approaches, as the resolution of digital elevation models (DEM) has typically been limited to 30–300 m (Saha et al., 2011; Iwahashi et al., 2018) thus inhibiting the detection of metre-scale features. Additionally, until recently,
sub–10 m datasets have been afflicted with patchy coverage (UKEA, 2023), limiting large–scale high–resolution digital mapping of smaller (sub–data resolution) landforms.

### 1.2 Machine learning as a solution

In recent years, automated landform mapping has taken a machine learning–based pattern recognition approach, starting with a DEM and extracting morphometrics from input data, such as slope, aspect, convexity and surface texture (Clubb
et al., 2019; Eisank et al., 2014; Saha et al., 2011; Iwahashi et al., 2018). Thus far, machine–learning approaches focused on large–scale geomorphological feature detection, such as sedimentary basins or terrain classification due to resolution restrictions (Kong et al., 2020; Iwahashi et al., 2018). Yet, the increasing availability of high–resolution (<10 m) datasets (e.g. Kartverket, 2021; UKEA, 2023) enable individual feature mapping for large areas. Norway has a national high–resolution (0.6 - 1 m) DEM and orthophotography data coverage (Kartverket, 2021). In conjunction with this, the development of new and
more robust machine learning methods, coupled with the effectiveness of older methods with new, high–resolution data (clustering and segmentation, Gentleman & Carey, 2008) suggest new approaches for small–scale landform detection.

As machine–learning is both fast, low on labour intensiveness once set–up and minimises human error (Gentleman & Carey, 2008), it is clear that it may be possible to address prior limitations to landform mapping. However, as high–resolution data availability is a recent development, only few studies, for example Corr et al. (2022), used a supervised random forest
(RF) algorithm to detect supraglacial lakes in Antarctica. Aydda et al. (2019) detected dune forms using three different unsupervised machine learning algorithms, including K-means clustering (KM). Here, we propose combining new, high–resolution data and machine–learning, to overcome previous limitations in geomorphological mapping.

### 1.3 Study aims

In this study, we develop a machine–learning algorithm to detect specific small–scale geomorphological landforms
in high-resolution DEM data, more specifically ribbed moraine. They usually have horizontal extents at a 10-100 m scale and are common in Norway (Dunlop & Clark, 2006; Dunlop et al., 2008; Hättestrand & Kleman, 1999; Finlayson & Bradwell, 2008). Furthermore, they typically form in shallow depressions in inland regions close to the former Fennoscandian ice divide (Sollid & Sørbel, 1994; Sarala, 2006; Fredin et al., 2013; Sommerkorn, 2020, Patton et al., 2016; 2017; Butcher, 2022). With this information, we can define potential study regions, and validation dataset. Ribbed moraines are subglacial ridges transverse
to the glacial flow direction (Dunlop & Clark, 2006), and were first documented as "Rogen moraine" by Lundqvist (1969). They also are known to form near morphologically similar landforms such as drumlins and hummocks (Lundqvist, 1989; 1997; Ely et al., 2016; Möller & Dowling, 2018), allowing us to test whether our approach can differentiate between different landforms of similar spatial extent. In addition, we note scientific interest in specifically mapping ribbed moraines, the origin of which is still a matter of debate (cf. Möller, 2006; Lindén et al., 2008; Boulton et al., 1987; Fisher & Shaw, 1992; Dunlop
et al., 2008; Sollid & Sørbel 1994).



Here, we develop and test two simple machine–learning algorithms: KM and RF (Gentleman & Carey, 2008) applied on new high-resolution datasets to design a computationally efficient, accurate and transferable method for automatically mapping small–scale landforms. We define our performance metrics in terms of: efficiency – the processing speed of the method; effectiveness – the accuracy of the method; and transferability – how well the method performs in different terrain types on a regional/ country–wide scale. We further discuss the potential extensions to detect other landforms such as eskers, drumlins, megaripples etc.

## 2 Methods

### 2.1 Machine learning

There are several machine learning approaches used previously in geomorphological research that may be useful for automatic feature detection. Many others exist, including U–net, a deep learning image segmentation method (Ronneberger et al., 2015), however, a main consideration of this study is to improve the time–efficiency of landform detection. Despite rapid segmentation once trained, U–net takes large amounts of time to train compared to random forest and K–means clustering (KM). Thus, we identify random forest and KM as two simple and lightweight methods that are used for similar image segmentation problems. The first and simplest approach is unsupervised machine learning (Gentleman & Carey, 2008). Unsupervised machine learning produces a segmented output from input data and does not require training data (a dataset designed to describe what the algorithm should look for). The algorithm identifies clusters based on similarity of data properties (Gentleman & Carey, 2008). One such method is KM, often used in image segmentation (Burney & Tariq, 2014). Secondly, supervised machine learning methods such as Random Forest (RF) require a training dataset, consisting of labelled data to provide a true reference for training the algorithm (Gentleman & Carey, 2008). RF has been used to great success for instance to map supraglacial lakes from satellite imagery (Discherl et al., 2020; Corr et al., 2022).

As ribbed moraines typically have spatial scales of 10s – 100s of metres, (Hättestrand & Kleman, 1999; Dunlop & Clark, 2006), we select both RF and KM as suitable candidate methods for detecting them in high-resolution Digital Elevation Model (DEM) data (Fig. 1). We will compare the output from the RF approach, the KM approach and a series of more complex algorithms. We will then compare the results to manually derived ground truth data (Fig. 1 b-d) to evaluate the performance of each method.

### 2.1.1 Random forest

The RF algorithm uses a series of decision trees (i.e. a forest) for classification, aiming to optimize agreement with training data (Gentleman & Carey, 2008). Within each decision tree, sequences of boolean questions subdivide the data (Breiman, 2001); the exact sequence and starting point of these questions is randomly altered for each tree of the forest. In this study, we test the RF method with 500 iterations using a majority vote method for final classification. Our RF method is trained on a small area of known ribbed moraines to the north of the Vinstre study area. Training data here has not been used in any other component of the study and is derived from ribbed moraines mapped by Sommerkorn (2020).



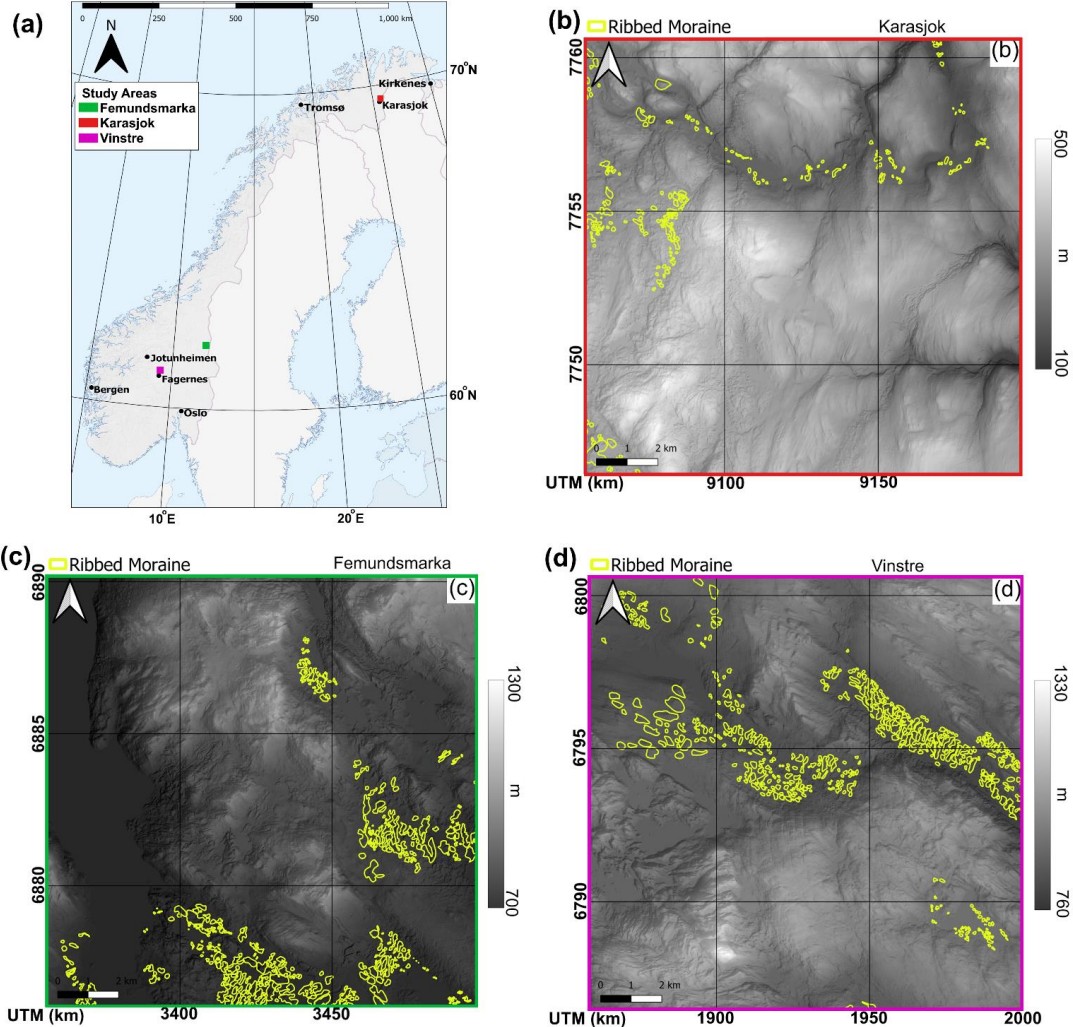

**Figure 1: (a) Map showing the location of the study areas throughout mainland Norway. Karasjok (b) is shown in blue, Femundsmarka (c) is shown in green and Vinstre (d) is shown in magenta. Each location shows a submap (b, c, d) displaying outlines of ribbed moraines (yellow) used as ground truth in this study (Sommerkorn, 2020). Additionally, we used an area bordering the north of the site displayed in (d) for training the supervised methods. Basemap made of Eurostat's GISCO administrative borders for Norway (GISCO, 2020). All maps projected in UTM 33N EPSG:5556, grid projection in (a) uses decimal degrees; b, c, d use UTM 33N m.**

### 2.1.2 K–means clustering

As an unsupervised algorithm, KM infers the defining parameters of each cluster based on the input data, and a series

of 'K centroid' points randomly scattered throughout the dataset. The algorithm clusters most alike points around centroids

through repeated iterations of the model (Fig. 2; Gentleman & Carey, 2008). KM is commonly applied in fields such as

photography and medicine (Burney & Tariq, 2014; Ng et al., 2020) but so far, has been rarely used in geomorphology (Lv et

al., 2013). The ideal number of K–centroids is determined after the first iteration of the model (Likas et al., 2003), by plotting

the sum of squared distances within the dataset against cluster count on a chart and selecting the 'elbow' point in the plotted

line (Marutho et al., 2018). After the defining of the optimal number of K–centroids, the model is ready for data output (Fig.

2).




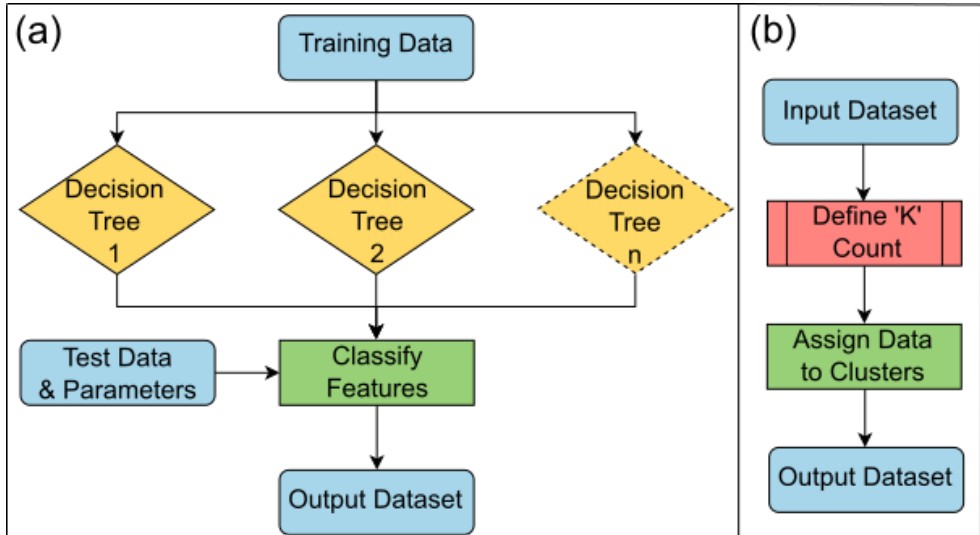

**Figure 2: Flowcharts describing (a) the basic principles behind the RF algorithm. (b) The basic principles behind the operation of a KM methodology. The model is first run to determine the number of K points required to represent the input data, and then run a second time to produce an output.**

### 2.1.3 Outline of composite methods

We use different combinations of the above–described algorithms, for three additional 'composite' methods. These are designed to determine whether the limitations of individual algorithms can be overcome by combining RF and KM, at the expense of time–efficiency. Combined approaches are common within machine learning studies, as they can often improve performance while avoiding micromanagement of the hyperparameters (Liu et al., 2021; Bhattacharjee et al., 2022).

The first composite method takes the output of a KM iteration to train the RF algorithm, rather than providing manually delineated training data. This method was selected as one of three combination methods with the aim of minimising subjectivity from outputs, allowing RF to determine features from an automated output, rather than a human–defined input. We term this method "KM-trained RF".

The second method is defined as the "OR" method, where both KM and RF are independently performed and the resulting classifications are merged by union, i.e. the final classification is positive if either KM or RF or both yield a positive. This method is designed to increase functionality if one or both methods are particularly strict on feature detection, or if each method functions more effectively in different areas.

The third method is defined as the "AND" method, where both KM and RF are independently performed and the resulting classifications are merged by intersection, i.e. the final classification is positive only if both KM or RF agree. We designed this method to increase functionality in situations where one or both methods have high rates of false positives.

### 2.2 Method application

### 2.2.1 Study areas

We selected three study areas representing different landscapes present in Norway for evaluation (Fig. 1). The landscapes included are high mountain (Vinstre), heath/moorland (Karasjok) and semi–mountainous marshland (Femundsmarka). Each study area covers 15 x 15 km, with data at a 1 m resolution (Kartverket, 2021). The first study area is south of lake Vinstre in central southern Norway (Fig. 1d). This region is representative for inland Norway, with a high mountain plateau interspersed with deep glacially formed valleys (Strøm, 1948). Furthermore, there exists high–quality ribbed



moraine mapping in and around the Vinstre study site (Fig. 3; Sollid & Sørbel, 1994; Sommerkorn, 2020), which we use for

training and validation of our methods. The second study area covers a subsection of the Femundsmarka National Park, near the Norway–Sweden border (Fig. 1c). This site is 30 km to the southwest of Lake Rogen, where ribbed moraines were initially identified and described (Lundqvist, 1969). A third study region lies in Finnmark, north of Karasjok, situated in a heathland and thus is largely different to the former two, both in form with large rolling hills (similar in morphology to ribbed moraines), and in geology, with much less exposed bedrock.

We trained each supervised method on a small 15 x 5 km subregion bordering the north of the Vinstre study area to determine the relevant parameters and morphometrics. We initially tested each method on the Vinstre study region, as the terrain in the Vinstre region is highly complex and typical of inland Norway, making it useful for ensuring the algorithm functioned well in similarly complex regions.

### 2.2.2 Map data

We use Norway's 1 m DEM and resample it to 10 m using bilinear interpolation for consistency instead of using an available 10 m product which was produced from various and uncertain sources (Fig 3.) (Kartverket, 2021b). The surface geology data is obtained as vector data from the Norwegian Geological Survey, NGU (NGU, 2022), and lake vector data from the Norwegian Water Resource and Energy Directorate, NVE (NVE, 2023)- We used nearest neighbour interpolation to convert these vector datasets into 10 m raster data, matching our resampled DEM. We obtained orthophotographs from Norway

in pictures, (NiB) (Kartverket, 2021a, c) and use this for ground–truth production and qualitative output validation. In addition, we use existing geological maps and mapped ribbed moraine ridges from Sommerkorn (2020).

### 2.2.2 Derived data

We use our raw data for two purposes: to produce training/testing datasets, and to generate morphometric and mask attributes for our machine learning algorithms. We manually delineated training/testing ground truth data using a combination

of previous moraine maps, the 1 m DEM and orthophotographs. We find that ribbed moraines in marsh areas are easily identifiable in orthophoto imagery due to their distinct contrast in colour to their surroundings, however in forests they are not detectable due to tree cover (Dunlop & Clark, 2006).

We derived morphometric and masking data from the DEM using several modules: the RichDEM python module (Horn, 1981; Zevenbergen & Thorne, 1987; Barnes, 2016), SAGA (Conrad et al., 2015) and the opencv python module

(Bradski, 2000). We also produced several filtered DEMs from the input 1 m DEM including: a 400 m low pass filter, a 30 m high pass filter, and a 30–400 m bandpass filter designed to isolate the size scale of ribbed moraines as defined by Dunlop and Clark, (2006). For our topographic wetness index (Boehner et al., 2002) data, we calculated indices using local values derived from the input DEM rather than global.

### 2.3 Morphometrics

We first select morphometrics based on the basic observable characteristics of ribbed moraines. In addition to the filtered elevation data (30 – 400 m; Hättestrand & Kleman, 1999; Dunlop & Clark, 2006) we generated: slope, general slope, curvature, planform curvature, profile curvature, spatial distribution (wavelength) and aspect. In addition, we derived the topographic wetness index (Boehner et al., 2002, Conrad et al., 2015) and topographic position index (Guisan et al., 1999) as metrics with the potential to add value to the segmentation algorithm. Each morphometric was identified from prior literature

(e.g. Eisank et al., 2014), and qualitatively tested for its ability to isolate ribbed moraine. With the combination of best-performing morphometrics we suggest the possibility of differentiating ribbed moraines from other landforms.

Secondly, we selected masking values, to remove as much noise from the input data as possible and avoid obvious misclassifications. This also aids our accuracy metrics by improving the balance of our data, as both RF and KM work most



successfully on balanced data, rather than unbalanced data (Pedregosa et al., 2011). Here, the best–balanced data is defined as

data where there is roughly a 50/50 split in the count of pixels per binary output category. Hence, masking values are used to remove as many pixels as possible where ribbed moraines do not or cannot form, balancing the dataset somewhat. From consulting literature, we defined areas where ribbed moraines cannot form: mountain peaks (Hättestrand & Kleman, 1999), steep slopes (Sommerkorn, 2020) and where surface bedrock is present (Sommerkorn, 2020). We also defined lakes as areas in which ribbed moraines are not detectable using DEM data. For each of these limitations, we produced a mask layer. Large–

scale (km<) peaks and ridges are masked from the dataset by setting a threshold on a laplacian of gaussian filter. The laplacian of gaussian is a second–order derivative edge detection filter, which identifies regions that show sudden steep changes in intensity (Kong et al., 2013). We mask steep slopes using a generalised slope layer generated from a low–pass filtered DEM (400 m <), slopes inclined greater than the thresholding value (e.g. Table 2) were masked out of the dataset. The low–pass filtered DEM is used together with a slope angle threshold to mask out the general valley slope angle. Additionally, we mask

mountains of a set wavelength from the testing to omit plateau–edge convex surfaces from analysis using a simple sinusoidal laplacian filter (Kong et al., 2013), referred to as a "mountain wavelength mask". In this case, mountains with a wavelength of greater than the threshold value were filtered out of the dataset. To mask out areas of exposed bedrock, we used national surface geological map of Norway (NGU, 2022) and mask all areas defined as exposed bedrock. Finally, we used NVE's lake database to mask out lakes (NVE, 2023).

Method outputs were visually compared to the locations of ribbed moraines as defined in Sommerkorn (2020). We made qualitative visual comparisons here as the changes made by different morphometrics were large, with numerical analysis coming as part of later predictor tuning. We made our visual comparisons against high–resolution orthophotography (Kartverket, 2021a, c), and Norway's 1 m resolution national DEM (Kartverket, 2021b). We then adjusted the algorithm to produce the most visually accurate output with the fewest inputs. Hence, in developing the method, we removed aspect as it

made no notable contribution to the output. We also conflated both planform and profile curvature into the total curvature metric, as we observed no difference in the output from any singular curvature parameter. In testing the value of the literature– defined morphometrics, we found topographic wetness index to be highly useful, but topographic position index to add no value to the output in any combination.

### 2.4 Workflow in practice

We implemented the series of segmentation algorithms on the combination of raw and derived data. We began by pre–processing the datasets to fit with each chosen study area, defined by the borders of the associated 1 m resolution DEM tile (Fig. 1). Once pre–processed, we ran each test algorithm on the datasets, for Vinstre, Femundsmarka and Karasjok. We also carried out a second iteration of RF using KM–produced training data, and combined outputs from KM and RF to produce each of our composite method outputs. Once we had completed each method, we prepared output datasets for statistical

analysis via a standard confusion matrix (Fig. 4).



**Figure 3: Illustration of morphometrics used in this study, with red line indicating the transect taken for subchart figures. (a) to (f) show clustering values, whilst (g) to (i) show masking values. (a) Shows elevation from the 1 m resampled to 10 m DEM data (Kartverket, 2021), (b) shows elevation with km-scale features filtered out, (c) shows the**

**output of the general slope (low frequency slope) filter, (d) shows the local slope values used for clustering, (e) shows curvature values used for clustering, (f) shows topographic wetness index (TWI), (g) shows masking values used for excluding regions where ribbed moraine cannot form, (h) shows curvature derived from a laplacian of gaussian filtered DEM for masking mountain peaks (i) shows a similar plot at 1 km resolution for excluding large wavelength features.**

## 2.5 Comparative Analysis

The first metric to test each method was efficiency – defined as the required CPU time. Then, we measured the

effectiveness of each method in detecting ribbed moraines. For each of the five methods, we derived the confusion matrix,





stating the numbers of true positive, true negative, false positive and false negative pixels, comparing model outputs to the manually delineated ground truth data (Fig. 5; Hong & Oh, 2021).



|  | Ribbed Moraine | Not Ribbed Moraine |
|---|---|---|
| Ribbed Moraine | True Positive | False Positive |
| Not Ribbed Moraine | False Negative | True Negative |

**Figure 4: Binary confusion matrix diagram showing actual vs. predicted values on a 2 x 2 grid.**

In addition to producing a visual representation of each method's effectiveness (Fig. 5), we calculated accuracy metrics: balanced accuracy (BA (Equation 4); Brodersen et al., 2010), and F–Score (Sokolova et al., 2006), each of which represents different sides of the confusion matrix, and each ranging between 0 and 1. We selected BA and F–score over a standard accuracy metric (Equation 1) as standard accuracy is skewed positively in unbalanced datasets. As our dataset

comprises regions where ribbed moraine do not make up 50% of the mapped area, our data can be considered unbalanced, hence we make use of BA. To calculate BA, we use Equation 4.

$$Accuracy = \frac{(True\ Positive + True\ Negative)}{(True\ Positive + False\ Negative + False\ Positive + True\ Negative)} \tag{1}$$

$$Recall = \frac{True\ Positive}{(True\ Positive + False\ Negative)} \tag{2}$$

$$Specificity = \frac{True\ Negative}{(True\ Negative + False\ Positive)} \tag{3}$$

$$BA = \frac{(Recall + Specificity)}{2} \tag{4}$$

BA accounts for the imbalance between classes, and it works effectively in identifying the influence that positives have on the accuracy of a dataset, hence a higher BA score represents a dataset with good positive returns versus Ground

Truth. F–score is computed as the harmonic mean of precision (Equation 5) and recall (Equation 6).

$$Precision = \frac{True\ Positive}{(True\ Positive + False\ Positive)} \tag{5}$$

$$F-score = 2 \cdot \left( \frac{(Precision \cdot Recall)}{(Precision + Recall)} \right) \tag{6}$$

F–score ignores true negatives, and like BA works well on imbalanced datasets. F–score typically will be higher when there are lower false positive and false negative, as it pays less attention to true positives and true negatives than BA. Therefore, in

working with both metrics, we gain a detailed understanding of the accuracy of our output.

**2.6 Final algorithm adjustments**

Once we identified which algorithm provided the best combination of effectiveness and efficiency, we improved the performance by adjusting the models' parameters. We took two approaches to make these adjustments: systematic and





intuitive. For systematic improvements, we adjust parameter values in small increments over a physically plausible range:

general slope mask threshold values are adjusted by 0.01 from 0.65 – 0.75, laplacian curvature mask threshold values are adjusted by 0.005 from 0.04 – 0.08, general slope kernel size is adjusted by 10 m from 300 m – 400 m, and mountain wavelength masking is adjusted by 100 m from 1 km to 2 km (Table 2). These tweaks thus provide an objective analysis of outputs based on pure accuracy scores.

For our intuitive approach, we changed inputs with the aim to maximize visual agreement between the classification

and the evaluation data. This approach served to health-check our dataset, ensuring that our method specifically detected ribbed moraines rather than "detection by accident". We used both approaches as a means of performing a robust sensitivity analysis, ensuring that specific landforms were detected. In certain situations, many pixels of moraines may be identified, but many conjoining pixels could also be detected as "ribbed moraine", thus only detecting ribbed moraine fields, rather than discrete landforms. Hence, while we aim to use the systematic analysis to determine the parameters with the highest statistical accuracy,

we used the intuitive analysis to determine the parameters which lead to the highest accuracy scores while still detecting discrete landforms rather than fields. The intuitive analysis involved changing the same parameters as in the systematic tweaks, whilst also changing cluster counts by ±1, and minor adjustments at the scale of 10 m to the filtering kernel size used on the DEM.

### 3.0 Results

### 3.1 Comparative method results

Our segmentation methods were quick to run (10 – 60 s) and produced reasonable outputs on a mid–range laptop (Intel i5–1135G7, 2.40GHz, 2420 MHz, 4 core. Intel Iris Xe Graphics. 16GB 3200MHz RAM). However, the outputs varied greatly between methods and study areas. As such, we separated our results between each segmentation algorithm and study area, and then outlined the mean results between each study area for each method.

### 3.1.1 Vinstre

In our test and evaluation of the five chosen methods at Vinstre, we find reasonable outputs for each method, showing the capabilities of detecting ribbed moraines in the terrain of this region (Fig. 5). However, each algorithm varies through its false positives and false negative detection rates. The standard accuracy (Equation 1) values (Table 1) show accuracy to be over 88% for all methods, with a mean of 0.926. However, due to the natural imbalance of our data, this metric is largely

dominated by the abundance of negatives. We expected some exaggerated false positive detection, yet some methods are better than others at omitting this. In addition, some of the methods detect more false positives than others, which is more difficult to exclude from the dataset. For example, we can see that KM and OR outputs show consistent false positives on the rounded peaks in the Vinstre study area's south; while the KM–trained RF algorithm shows a widespread pattern of random false positive detection (Table 1). The RF and AND methods yield fewer positive pixels overall, which leads to lower accuracy

scores as only the banks or ridges of ribbed moraines are detected, rather than full features.

Statistically, we find the KM and OR methods to have the highest and most consistent accuracy metrics, suggesting that these methods have the most use in ribbed moraine detection. We see this with KM returning BA of 0.75 σ (standard deviations) above the mean, and F–scores of 0.89 σ above the mean (Table 1). The OR output echoes this, with BA and F values of 1.16 and 1.00 σ above the mean respectively (Table 1, Fig. 5). On the other hand, while the AND approach is the

second most effective method in terms of overall accuracy, we see poor results in balanced metrics, giving values of 1.20 σ below the mean value. Furthermore, the KM–trained RF method shows the least promising visual output, the least consistent and near–least promising accuracy output, where accuracy = 1.34 σ below the mean, BA = 0.57 σ below the mean, and F–





score = 0.41 σ above the mean (Table 1). Therefore, as an overall ranking, we rate the OR method as the best performing in the Vinstre region, while the KM–trained RF method returns the least successful results.

**3.1.2 Femundsmarka**

On comparing results from Femundsmarka, we find a similar pattern to Vinstre, with high false positive and negatives common between KM–trained RF. Additionally, RF and AND classifiers, showed distinct features in qualitative detection. False positive detection of rounded hills in the Femundsmarka region is reminiscent of the Vinstre region in the KM and OR classifiers. We also rank each method the same as at the Vinstre site, whilst also showing that the variation in performance

from the mean is similar between both sites. We find major differences are only present within the KM–trained RF method and the RF method in terms of variability. For these, KM–trained RF returns values of 1.27, 0.55 and 0.25 σ below the mean for Accuracy, F-score and BA, respectively. RF on the other hand, returns 1.04, 0.09 and 0.79 σ above the mean. In this case, we see that the KM–trained RF method consistently trended towards below-mean performance, whereas RF consistently performs above-average.

**Table 1: Performance scores for each study area (Fig. 5); metrics are rounded to 3 decimals. As BA and F-score are combinations of recall and specificity, (Equations 2 and 3) we only include the BA and F-score outputs in this table.**

**Vinstre**

| METHOD | Accuracy | F–Score | BA |
|---|---|---|---|
| KM | 0.915 | 0.229 | 0.679 |
| RF | 0.963 | 0.163 | 0.550 |
| KM+RF | 0.875 | 0.147 | 0.653 |
| OR | 0.913 | 0.244 | 0.698 |
| AND | 0.965 | 0.112 | 0.531 |
| *Average* | *0.926* | *0.179* | *0.622* |
| *Std_Dev* | *0.038* | *0.056* | *0.076* |

**Femundsmarka**

| METHOD | Accuracy | F–Score | BA |
|---|---|---|---|
| KM | 0.925 | 0.214 | 0.639 |
| RF | 0.963 | 0.182 | 0.556 |
| KM+RF | 0.903 | 0.148 | 0.614 |
| OR | 0.925 | 0.238 | 0.661 |
| AND | 0.964 | 0.104 | 0.529 |
| *Average* | *0.936* | *0.177* | *0.600* |
| *Std_Dev* | *0.026* | *0.053* | *0.056* |

**Karasjok**

| METHOD | Accuracy | F–Score | BA |
|---|---|---|---|
| KM | 0.881 | 0.059 | 0.666 |
| RF | 0.981 | 0.135 | 0.566 |
| KM+RF | 0.824 | 0.029 | 0.617 |
| OR | 0.881 | 0.059 | 0.666 |
| AND | 0.984 | 0.091 | 0.536 |
| *Average* | *0.910* | *0.075* | *0.611* |
| *Std_Dev* | *0.070* | *0.040* | *0.059* |

**Averages**

| METHOD | Accuracy | F–Score | BA | Time (s) |
|---|---|---|---|---|
| KM | 0.907 | 0.168 | 0.662 | 18 |
| RF | 0.969 | 0.160 | 0.558 | 1085 |
| KM+RF | 0.867 | 0.108 | 0.628 | 1103 |
| OR | 0.906 | 0.181 | 0.675 | 1433 |
| AND | 0.971 | 0.102 | 0.532 | 1493 |
| *Average* | *0.924* | *0.144* | *0.611* | *1026* |
| *Std_Dev* | *0.045* | *0.036* | *0.063* | *594* |



### 3.1.3 Karasjok

Performance patterns in the Karasjok region depart from those in the previous areas. There is greater variability
between algorithms and a general increase in misclassifications. The RF and AND methods perform well in Femundsmarka
and Vinstre, while the KM, KM–trained RF and OR methods produce outputs with speckled pixels. Many false positives are
present, particularly in the southern part of the Karasjok region. It is likely that these are a result of different landscape
morphologies between Vinstre, Femundsmarka and Karasjok, where the latter is a moorland environment, versus the prior
two, which are more typical of inland Norway (Hjort et al., 2014). Statistically, we also see poor performance KM rates below
the mean standard accuracy and F–score metrics by 0.40 σ, whilst maintaining a high BA score of 0.93 σ above average (Table
1). In short, this means KM is effective at detecting true positives but classifies too many false positives. RF is opposed to this,
with accuracy and F–score returning values of 1.01 and 1.50 σ above average respectively, versus 0.76 σ below average for
BA. The OR method returns identical values to KM, as RF under–classifies features in the Karasjok region, thus allowing KM
to dominate the OR output in this area.

Despite the differences in the Karasjok region, we still see similarities between the Karasjok region and the prior two,
with both the KM–trained RF and AND approaches having inconsistent variability from the mean, suggesting their lack of
value as methods. Due to the locality of the false detections found in the KM and OR methods, it is possible that they are
detecting features common to this region, which likely have a similar form to ribbed moraines found elsewhere in Norway.

### 3.1.4 Overall performance

Studying the overall performances of each method across the different study sites, we find similar results in the first
two regions, where KM ranks consistently second in BA, and OR ranks highest. KM performance scores are both above
average by 0.38 σ and 0.81 σ, while all other methods show some inconsistency between accuracy metrics. For example, the
OR method ranks 1.02–1.03 σ above the mean BA but ranks 0.40 σ below the mean performance. Hence, while KM does not
rank as highly as OR in BA, its output is more consistently accurate. As a result, we can discount the combined, RF and AND
methods, as they all consistently rate third or worse in accuracy metrics, showing their lack of segmentation effectiveness.

In addition to accuracy, we determined the CPU time required for each method (Table 1) as a measure of efficiency.
On a mid-range laptop (Intel i5–1135G7, 2.40GHz, 2420 MHz, 4 core. Intel Iris Xe Graphics. 16GB 3200MHz RAM), KM
takes only 18 seconds on average per iteration, 1.67 σ faster than the mean rate, while every other approach takes two orders
of magnitude more, at over 1000 seconds. When considering our two most statistically effective methods, we find KM is
notably faster than the OR methodology. Due to its much lower computational cost, we selected KM as the method of choice,
despite its slightly lower performance.

### 3.2 K–means refinement

Upon producing a baseline set of predictors for the KM methodology, we conducted many predictor adjustments as
a sensitivity analysis. These values aided in determining two final "best–fit" methods based on: statistics through a systematic
approach; and statistics combined with qualitative observations made on the output maps.



**Figure 5: Comparison of input to output in the three study regions. (a, c, e) show ground truth data, while (b, d, f) show accuracy maps denoting confusion matrix values for the clustered output of the KM method. Outputs are superimposed on a hillshade of the input DEMs (Kartverket, 2021b).**





### 3.2.1 Systematic analysis

The systematic analysis yields optimal predictor values, in terms of average BA scores for each iteration per region.
For example, the optimal value of the general slope threshold is 0.71, where BA values are 0.76, 0.61 and 0.72 for the Vinstre, Femundsmarka and Karasjok sites respectively. The optimum BA scores 1.7σ above average for the range of tested values (Table 2).

In addition to our best outputs, we note interesting values and themes throughout the systematic analysis. In general, we found each region to have similar patterns of maximal performance. The only commonality between our three sites under
the general slope kernel value tweaks was with a parameter value of 300 m (where the kernel size for slope smoothing is 300 m resolution), where all scores were above 0.64 BA. The pattern of scores calculated using 300 m general slope kernel, (0.74 Vinstre, 0.65 Femundsmarka, 0.73 Karasjok) is common throughout this parameter's output. While Femundsmarka generally follows the pattern of values in the other two datasets, changes are more muted, with values ranging between 0.61 and 0.67 for general slope kernel, general slope threshold and laplacian curvature threshold, versus changes of double this magnitude
in the Vinstre and Karasjok regions. Furthermore, while there is general agreement throughout the data, Vinstre appears to be at odds in optimal values to Karasjok for accuracy scores in laplacian curvature threshold tests, with optimal values having no relation.

**Table 2: Table showing predictor values from systematic analysis with BA as the key accuracy score. BA Range shows the range from minimum to maximum values of BA output. Each predictor was tested on an individual basis.**

|  | Location | Value | Value Range | BA Global Average | BA Range | BA Standard Deviation |
|---|---|---|---|---|---|---|
| **General Slope Threshold** | *Vinstre* | 0.71 | 0.65 – 0.75 | 0.71 | 0.606 – 0.757 | 1.7 σ above mean |
|  | *Femundsmarka* | 0.71 |  |  |  |  |
|  | *Karasjok* | 0.71 |  |  |  |  |
| **Laplacian Curvature Threshold** | *Vinstre* | 0.07 | 0.04 – 0.08 | 0.67 | 0.608 – 0.745 | 0.9 σ above mean |
|  | *Femundsmarka* | 0.07 |  |  |  |  |
|  | *Karasjok* | 0.07 |  |  |  |  |
| **General Slope Kernel** | *Vinstre* | 300 m | 0.3 km – 0.4 km | 0.71 | 0.601 – 0.774 | 1.8 σ above mean |
|  | *Femundsmarka* | 300 m |  |  |  |  |
|  | *Karasjok* | 300 m |  |  |  |  |
| **Mountain Mask** | *Vinstre* | 1000 m | 1 km – 2 km | 0.76 | 0.538 – 0.791 | 2.4 σ above mean |
|  | *Femundsmarka* | 1000 m |  |  |  |  |
|  | *Karasjok* | 1000 m |  |  |  |  |

The tweaks made to the mountain wavelength mask present much more varied results than the other tests; but a consistent optimal or near–optimal value. This lack of consistency is likely due to the different wavelengths of mountains in each study area. But additionally, it appears that a consistent wavelength for mountain features is roughly 1 km between trough and peak. Our results clearly show this, with maximum BA values of 0.79 (Vinstre), 0.76 (Femundsmarka) and 0.73 (Karasjok).
This high performance also mean that the mountain mask parameter is the most influential parameter on statistical detection, improving the mean BA across the board by 0.10 (3.6 σ above the mean improvement). On the other hand, the least influential parameter is laplacian curvature threshold, with an improvement in accuracy of only 0.03, only 0.9 σ above the mean BA score. This result, however, is consistent with the lack of agreement between datasets when tweaking the laplacian curvature
threshold value.

### 3.2.2 Intuitive approach

Results from our intuitive analysis show several outputs with high BA. Each of these approaches yields BA scores > 0.70 at Vinstre, and a mean of > 0.69 across all regions. Yet, visual evaluation reveals that settings in which BA score is > 0.70 can lead to moraine overclassification (Fig. 5). Thus, despite high BA, this validates our health-check, demonstrating that




higher statistical performance does not necessarily produce the best performing output for discrete feature detection. This potentially comes from the complexity of the terrain in which we iterate the method, with areas of higher relief generally performing worse than lower relief when using geomorphometric methods (Hjort et al., 2014). In short: we find BA < 0.70 leads to discrete detection of ribbed moraines, whilst BA > 0.70 delineates ribbed moraine fields, filling the gaps between features. Hence, we outline two potential approaches: (1) That high accuracy should be the focus, aiming to detect fields of

ribbed moraines as in previous studies, and (2) that discrete ribbed moraines should be the focus, aiming to detect individual ribbed moraines on a large scale.

We also note several patterns between our analysis approaches. Firstly, we find less agreement between all three regions in the statistical output of our trial–and–error intuitive approach versus the systematic approach. Despite this, we also find there is consistent agreement in our most effective discrete landform detection method, with scores varying by < 1 σ.

Interestingly, we find that again, Femundsmarka shows the lowest variability in BA score, with a standard deviation of 0.29 σ, versus a standard deviation of 0.50 σ for Vinstre, and 0.56 σ for Femundsmarka.

## 4 Discussion

### 4.1 Pre–predictor tuning

The high level of agreement with ground truth in the Vinstre region show the KM algorithm's potential. Through F–

score and BA, we find the KM method to be most successful in the Vinstre study area, likely due to the method being originally initiated on the complex, mountainous terrain of inland Norway. This is promising because it shows the method performs well at detecting ribbed moraine in regions of high relief, one of the most common areas for ribbed moraine to form (Sollid & Sørbel, 1994; Sommerkorn, 2020). Hence, we assert this method is transferrable throughout central Norway and other similarly mountainous regions. Yet, we find the main shortcoming of the pre–tuning method in the Vinstre study region to be

overdetection of riverbanks, palaeochannels and other, morphologically similar landforms. This is potentially in-part due to the interpolation of 1 m to 10 m data averaging morphometric signals across small-scale landforms.

We additionally find good performance in the Femundsmarka study area, with an F–score of 0.21. This is most likely due to the partial similarity to the Vinstre study area, with the presence of some high mountain terrain. However, complexity in the local relief increases, including three terrain types: moorland, lake and high mountain, versus the consistent high

mountain terrain of the Vinstre study area. While lakes are masked out, the mixed semi-mountainous marshland of Femundsmarka likely contributes to the lower accuracy scores due to transferability issues between different reliefs (Hjort et al., 2014), as Femundsmarka's results show the algorithm attempting to classify lakeshore and rounded moorland hills as ribbed moraine (Fig. 5d). This reflects the riverbank detection in Vinstre, and thus is most likely due to the similarity of morphology, particularly in relation to slope and curvature (Dunlop & Clark, 2006; Lindén et al., 2008; Möller, 2006).

Additionally, Femundsmarka shows greater rates of false negatives than Vinstre, with many ribbed moraines in the south of the study area (Fig. 5d) being unclassified. The cause of the false negatives is unclear, but they could be attributed to the lower overall difference in elevation between ribbed moraines and their surroundings (30-40 m North, 20-25 m South) in southern Femundsmarka. Despite some of the limitations, we consider our initial results from Femundsmarka promising.

With an F–score of 0.06, Karasjok has a greater rate of false positives than other regions (fig. 5). This may be a result

of the landscape being predominantly comprised of rounded hills of similar appearance to ribbed moraine, similarly to areas of Femundsmarka. BA is 0.67, in comparison to the 0.64 and 0.68 for Femundsmarka and Vinstre, respectively. Hence, the primary issue with the Karasjok region is overdetection (increased false positives). We propose the issues of false positive detection we observe in Karasjok to be a result of inherent transferability issues that arise in geomorphometry. Hjort et al. (2014) describes the difficulty in extrapolating geomorphometric methods from high to low relief areas, as region specific

conditions can have a strong impact on the relative importance of different variables. In the case of Karasjok, the change in



relief likely leads to a greater influence of the curvature metric. Alternatively, this could be a product of the interpolation between 1 m and 10 m leading to an inaccurate curvature value for small scale channel features.

The overdetection we observe in our study regions could be limited by predictor tuning (Moradi Fard et al., 2020) or through classifying detected features post–detection. For this study, we choose to take a consistent approach and undergo predictor tuning to improve our methodology, due to the initially promising output from our approach.

## 4.2 Finalised outputs

### 4.2.1 Computation considerations

For all iterations, we found that the relatively light computing load opens the possibility to apply the method to larger scales such as mapping moraines all over Norway. In addition, the parallelized implementation of K–means named "minibatchkmeans" (used in this study) (Pedregosa et al., 2011) further increases the processing speed on multicore servers.

### 4.2.2 Performance

In computing BA, we find reasonable (0.59) to good (0.78) scores depending on iteration and location. Mean BA scores vary by 0.11 (0.60 to 0.71), which we expect given the varied terrain types. This outlines issues in transferring between terrain types and maintaining performance. For example, during our systematic analysis, we set the general slope threshold to 0.75, and find BA values of 0.63 (Vinstre), 0.62 (Femundsmarka) and 0.73 (Karasjok), demonstrating better performance in the Karasjok moorland than in high mountain terrain. Due to discrepancies between what can be considered valuable in a qualitative versus quantitative results, BAs below 0.75 do not suggest poor method performance, as values between 0.65 to 0.70 show the greatest performance in this study for detecting discrete moraines. It is, therefore, important to consider that the algorithm is not designed to detect ribbed moraine but "ribbed moraine–like" features. This means there will always be a degree of over/under classification, particularly in complex post–glacial and real–world landscapes due to the variability in ribbed moraine shape (Dunlop & Clark, 2006). We will also see over/under classification due to geomorphological features of similar shape within our study areas, as ribbed moraines do not develop in isolation (e.g. drumlins, channels, mega–scale glaciolineations; Ely et al., 2016). Hence, over/underdetection can be perceived positively, as output features can be classified post–detection using parameters such as orientation versus glacial streamflow, with ribbed moraines as perpendicular to, and drumlinoid forms streamlined with flow directions (Dunlop & Clark, 2006; Ely et al., 2016).

Secondly, we find that BAs below 0.70 may be of greater value than those above 0.70 if we are aiming to identify discrete features. Our results show that BAs of more than 0.70 present more true positives, resulting in good statistical results, but many more false positives are also present, as spaces between ribbed moraines are detected as ribbed moraine. This results in areas between ribbed moraine being filled up, leading to positive detection of moraine features, but discrete moraines are poorly captured Hence, we consider a range of results as successful: BA 0.65 to 0.75 depending on whether the aim is to classify fields or discrete landforms. This is due to BA alone not being the ideal parameter for measuring the value of our output – but instead we require complementary assessment, through qualitative analysis as a means of determining output value, on top of the ideal range of BA values. We thus believe both sets of BA scores are reasonably accurate, particularly as our raw accuracy never drops below 0.82, representing an 82% per–pixel success rate.

### 4.2.3 Transferability

The transferability of this algorithm depends on one major component: the wide-scale landscape type and relief, showing similar issues with transferability to previous works (Hjort et al., 2014). We find the method works well when applied to similar landscape types, as seen in Femundsmarka and Vinstre. When transferred to different landscape types and reliefs, (i.e. between high mountain, moorland, and mixed marsh-mountain terrain) we observe poorer detection of ribbed moraines. As a result, this algorithm is transferable for most landscapes where ribbed moraines occur in inland Norway (high mountain),



however, for regions exhibiting differences to the high mountain regions, such as moorland and coastal regions, we can tune the model accordingly at the scale of each DEM tile. This is relevant when considering expansion to other regions of the world, as much of the Canadian shield is landscape close in format to Karasjok – which would indicate a need for predictor tuning (Dunlop & Clark, 2006; Dunlop et al., 2008).

**4.2.4 Detection of new features**

While analysing our outputs, we found evidence of ribbed moraines detected by the algorithm that were not present in ground truth data. These were commonly found in the Femundsmarka region, in areas adjacent to manually marked ribbed moraine (Fig. 6). The detected landforms occur in small clusters, with a similar pattern to ribbed moraine as described in Dunlop & Clark (2006). We suggest that this may be due to a series of factors, including the presence of forest and farmland,
obscuring ribbed moraine in stereo–imagery or poor DEM data used for manual production of ground truth data. Hence, we suggest that while the method has some limitations, there are also clear advantages as the method uses LiDAR data, which does not exhibit the same issues in differentiating between landforms under forest cover as spectral imagery.

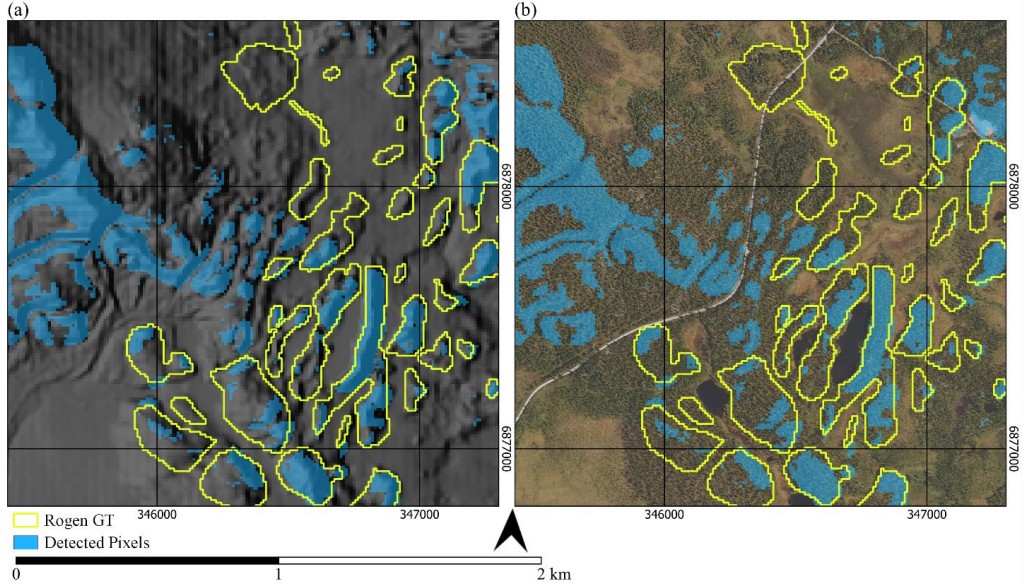

**Figure 6: Comparison of ground truth to detected pixels on (a) hillshade and (b) Orthophoto imagery (Kartverket, 2021a, c). Here, GT refers to "Ground Truth".**



### 4.2.5 LGM ice flow direction proof of concept

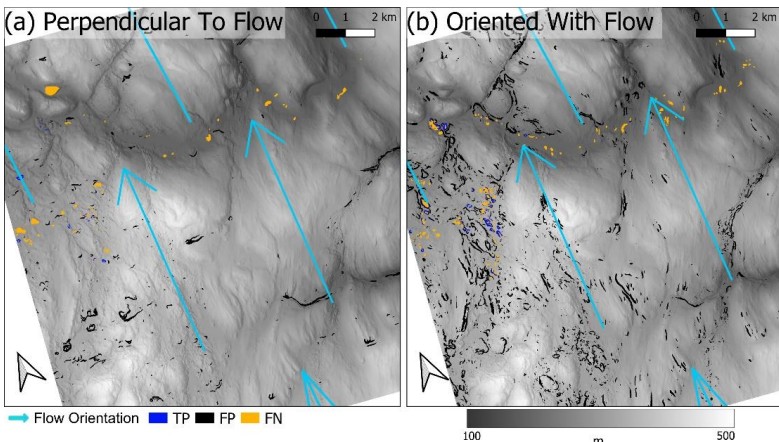

**Figure 7: Detected features versus LGM ice flow direction in the Karasjok region. (a) shows values ±25° from orientation perpendicular to mean flow direction (blue arrows) during the FIS (Fennoscandian Ice Sheet) (Patton et al., 2016; 2017) and (b) shows values not oriented perpendicular to average flow angle during FIS (Patton et al., 2016; 2017). Both figures superimposed on 10 m DEM and hillshade (Kartverket, 2021b).**

To explore possibilities for further refinement, we calculated the orientation of each detected polygon using a simple bounding box method, and determined the orientation of its longest axis. We compare these orientations with modelled flow direction of the FIS by comparing feature orientation to the mean flow direction of the ice sheet between 11kya and 25kya (Patton et al., 2016; 2017). Fig. 7 shows polygons perpendicular or streamlined (± 25 °) to glacial flow, helpful in distinguishing landforms oriented along (drumlinoid) from those oriented transverse (ribbed moraines) to flow direction (. Yet, we note limitations with hummock forms (Ely et al., 2016), as they have no dominant orientation and could end up in any group, demonstrating a need for a more robust classification method, potentially based on feature aspect ratio. We also note this method has challenges detecting the orientation of anastomosing features, a common type of ribbed moraine (Dunlop & Clark, 2006). Another potential limitation of this method is the low resolution of modelled glacial flow which is in the order of kilometres, wider than many glacial valleys.

### 4.3 Avenues for method improvement

We identified two areas of improvements. First, we suggest improving automation, which would allow scaling to larger areas. For example, through inclusion of automated predictor tuning for landscape types in the model parameters (parameter profiles). Automation would allow for application on a country scale with minimal input through the implementation of these parameter profiles, minimising false positive and false negative identification in varied landscapes. We envision four landscape types for Norway: high mountain (our original approach), low mountain, moorland (Karasjok) and coastal. We believe these four categories would cover the main landscape types common across regions where ribbed moraines are present. These categories would likely prove sufficient in the regions outside Norway where ribbed moraines are known to exist (e.g. Canada, Ireland, United Kingdom, Sweden, Finland), however regional differences in scale and geology may require landscape type parameterisation.

Second, we identify the importance of implementing a post–detection classification of landforms. Our results show that overclassification is common, particularly as there are many landforms with similar geomorphological properties to ribbed moraines (Fig. 5). We suggest future works include investigation into landform classification, as it would allow for identifying specific landforms including drumlinoid forms, ribbed moraine and others, improving the value of the methodology. This would potentially aid the continuum hypothesis of Ely et al. (2016), by showing landform transitions from ribbed moraine. Hence, we consider this as the next logical step in method development. In addition, we suggest that with post–detection



classification, this method could be used on a wider basis than only ribbed moraine, due to its ability to easily detect specific
geomorphologies based on relatively basic morphometric traits.

**5 Conclusions**

To develop a method for automated mapping of landforms throughout Norway, we tested two machine learning
algorithms and three composite approaches. We determined ribbed moraine as a suitable example landform, which would also
be scientifically interesting to identify. Through testing, we settled on using an unsupervised K–means clustering algorithm
for moraine detection, thus requiring no training data. Ground truth data for our testing was produced through manual analysis
of high–resolution elevation model (Kartverket, 2021) derivatives, and high–resolution aerial/ satellite imagery (Kartverket,
2021).

Our results demonstrate unsupervised machine learning as sufficient for the automated detection of geomorphological
features through a simple KM approach, rather than the need for complex supervised machine learning methods. We also
demonstrate that minimal data are needed for this approach, with only high–resolution DEM derivatives, superficial geology
and a lake mask required for our methodology to function. We evidence this in initial testing, where a supervised RF method
averaged poorer performance than an unsupervised KM approach. In addition, we find that we can differentiate between
identification of discrete and fields of landforms. Hence, we find this method to be scalable, in that different output resolutions
are possible.

This study indicates the value of automated machine learning for landform detection as a means of minimising time
spent delineating features manually and in–field. We detect ribbed moraines throughout our study areas in relation to ground
truth data and detect a small number of previously unmapped ribbed moraines in one of these. Thus, we show the value of an
objective and systematic method using new, high-resolution data in detecting features. Furthermore, we detect many additional
features in all study areas, which have similar morphologies to ribbed moraine but powerful refinement can be achieved by
considering orientation with respect to the former ice flow direction.

In summary, this study demonstrates that unsupervised machine learning is a viable and efficient method for the
automated detection of ribbed moraines and similar features based on modern high–resolution DEM (Kartverket, 2021b). With
our promising results, we identify a future path for such methodologies in geomorphology, as a means of updating
geomorphological maps and producing new geomorphological maps where we have high-resolution data to input (e.g. dune
mapping on Mars). Thus, we intend to develop this work into mapping ribbed moraine and adjacent features throughout
Norway, so as to aid in developing our understanding of the geological history of Fennoscandia.

**Code availability statement**

Finalised code is available in the repository Aeteia/Ribbed–Moraine at https://doi.org/10.5281/zenodo.7991094, (Barnes &
Filhol, 2023).

**Data availability statement**

The supporting data produced for this paper are openly available in the repository Aeteia/Ribbed-Moraine at
https://doi.org/10.5281/zenodo.7991094, (Barnes & Filhol, 2023)**.** The input data for this paper are freely available and can be
found in the Norwegian national geospatial data archive at the following links: NiB (Kartverket, 2021a, c)
https://kartkatalog.geonorge.no/metadata/norge-i-bilder/e7cd5f9b-20e1-4f59-b379-64828cd38062, national 1 m DEM
(Kartverket, 2021b) https://kartkatalog.geonorge.no/metadata/hoeyde-dtm1/0442c622-6639-4024-86be-846de8c15bb2, lake
database (NVE, 2023) https://kartkatalog.geonorge.no/metadata/innsjoedatabase/823b8639-9a49-41bf-8571-3608435eb149,



surface geology map (NGU, 2022) https://kartkatalog.geonorge.no/metadata/loesmasser/3de4ddf6-d6b8-4398-8222-f5c47791a757, Administrative borders (GISCO, 2020) https://gisco-services.ec.europa.eu/distribution/v2/countries/.

**Author contributions**

T. Barnes was the primary contributor to this paper in terms of each section of the CRediT guidelines. S. Filhol provided significant support in methodological development and refinement, including programming support and conceptualisation of method development – further S. Filhol provided review and editing support. T. V. Schuler provided conceptualisation and review / editing support, with methodological development guidance and project supervision. K. S. Lilleøren provided review / editing support, project supervision and methodology conceptualisation.

**Competing interests**

The authors declare no competing interests in the production of this manuscript.

**Acknowledgements**

We acknowledge L. S. Schmidt for her assistance in discussing possible ways of implementing our ideas in a programming sense, and computational support. We also acknowledge E. M. Lund for spellchecking the document.

**Financial support**

This study has been funded by Universitetet i Oslo. S. Filhol was funded from ClimaLand, an EEA/EU collaboration grant between Norway and Romania 2014-2021, project code RO-NO-2019-0415,1290 contract no. 30/2020.






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
