# Peer review of "A machine learning approach to the geomorphometric detection of ribbed moraines in Norway"

_EGUsphere, 2023_

## Referee Comment (RC1)

1. Does the paper address relevant scientific questions within the scope of

   ESurf?  Yes

2. Does the paper present novel concepts, ideas, tools, or data?

   Yes, this is clearly a novel approach to trying to automate the mapping of ribbed moraines, which cover extremely large areas of glacial terrain and such an approach would greatly reduce the time involved in manual mapping and could improve our understanding of ice sheet histories.

3. Are substantial conclusions reached?

   Yes, it is clear that the method produces good initial results that will allow the continuation of improvements to the mapping method.

4. Are the scientific methods and assumptions valid and clearly outlined?

   Yes, this is all clearly described in the paper

5. Are the results sufficient to support the interpretations and conclusions?

   Yes, the results are presented clearly and honestly and there is a good critical discussion on the shortcomings of the approach and ways this could be improved.

6. Is the description of experiments and calculations sufficiently complete and precise to allow their reproduction by fellow scientists (traceability of results)?

   Yes

7. Do the authors give proper credit to related work and clearly indicate their own new/original contribution?

   Yes

8. Does the title clearly reflect the contents of the paper?

   Yes

9. Does the abstract provide a concise and complete summary?

   Yes

10. Is the overall presentation well structured and clear?  Yes

11. Is the language fluent and precise? yes

12. Are mathematical formulae, symbols, abbreviations, and units correctly defined and used? Yes

13. Should any parts of the paper (text, formulae, figures, tables) be clarified, reduced, combined, or eliminated?   I have suggested the addition of one figure

14. Are the number and quality of references appropriate? They are generally good, but I have made suggestions for additional ones to be included.

15. Is the amount and quality of supplementary material appropriate? yes

General comments

This is a very good contribution to geomorphological mapping of glacial terrain and in particular ribbed moraine which automated mapping approaches to date have overlooked. Whilst it is clearly not perfect, which is freely acknowledged by the authors, it also clearly represents a great step in the right direction towards developing a mapping approach is in my opinion badly needed, because mapping ribbed moraines over the large areas needed to test formational theories and reconstruct ice sheet sectors is extremely time consuming work. This approach, at least shows good results in areas where the terrain cover is mountainous and should work well in Norway and parts of Sweden.

Specific comments and issues.

compact listing of purely technical corrections at the very end ("technical corrections": typing errors, etc.).

**Line 54.**  Is this sentence complete? 'In conjunction with this, the development of new and more robust machine learning methods, coupled with the effectiveness of older methods with new, high–resolution data  (clustering and segmentation, Gentleman & Carey, 2008) suggest new approaches for small–scale landform detection'

Should it not end in something like   'suggest new approaches for small–scale landform detection IS NOW POSSIBLE'?

**Line 70.** The reference to Lundvist in this line is not factually correct and you need to update your references on this see this from opening paragraph of Dunlop and Clark 2006 in QSR:

*Ribbed moraine, also known as Rogen moraine, are subglacially formed transverse ridges that cover extensive areas of the beds of the former Laurentide, Fennoscandian and Irish ice sheets (Fig. 1) and likely exist beneath the Antarctic ice sheet. Originally these landforms were thought to be created at ice margins (Frödin, 1954; Cowan, 1968) hence the use of the word moraine; but it is now widely recognised that they formed subglacially (Hoppe, 1952; Lundqvist, 1969; Aylsworth and Shilts, 1989; Bouchard, 1989; Hättestrand and Kleman, 1999).*

I suggest you update this text and include these references from this 2006 paper.

**Line 71.** This line needs to be rephrased….you state that '*They also are known to form near morphologically similar landforms such as drumlins and hummocks*' They as in ribbed moraine don't not form anything, they are the geomorphological signature of an ice sheet process so it is the formational process that forms similar landforms and drumlins and hummock, so rephrase so it is clearer. Also, you should reference that some authors argue that this suggests that the similarities and transitions from ribs to drumlins suggests there is a bedform continuum and you should reference this. I think it was Aario who first argued this, see this paper:

RISTO AARIO (1977) Classification and terminology of morainic landforms in Finland, Boreas
**https://doi.org/10.1111/j.1502-3885.1977.tb00338.x**

But there is evidence in Dunlop and Clark (2006) too for their being a bedform continuum from ribs to drumlins

**Line 72** states 'allowing us to test whether our approach can differentiate between different

landforms of similar spatial extent' what does this mean, I think you need to clarify this because you should explain how does your mapping approach test this?

**Line 79.** Change the term small-scale landforms to ribbed moraine, as this is what you are mapping and you clarify this by the time the reader gets to here. I suggest doing this throughout.

**Line 95.** I think you could include a few more references here and other examples of how Random Forest has been used in glacial mapping, for example the 2023 paper by Ali et al, in Journal of Glaciology where they mapped glacier change over large parts of the Arctic and high

latitudes https://www.cambridge.org/core/journals/journal-of-glaciology/article/glacier-area-changes-in-novaya-zemlya-from-198689-to-201921-using-objectbased-image-analysis-in-google-earth-engine/0225F790B915D57EBD4EAF76D5C1EE07

**Line 99.** Fig 1. I think it would be best to include at least one close up view of what these ribbed moraine look like, you cannot see any, only the mapping outlines. A new figure showing clear unmapped examples is needed in my opinion to show them clearly in the data. You can consider a new composite figure, with the unmapped ribbed moraine, next to the same ones mapped but in a more close up view, this would be a separate and new figure to your current figure one.

**Line 100**. Change the future tense from 'We will then compare' to 'We then compare'

**Line 161.** You need to identify what the relevant parameters and morphometrics, what are they?

**Line 165.** In section 1.2 of the paper you discuss a lot about the lack of progression in this field because the high resolution DEM data is not available, then you have the amazing countrywide 1m DEM that overcomes the issue for your study, but it is not clear why you are resampling Norway's 1 m DEM to 10 m making it more course. What is the point of doing this, you need to justify this in your methods as this must impact on the results in some way.

**Line 225**. Your figure numbering is wrong here, this should be Figure 3 not 4. This has had a knock on affect on the numbering of figures after this (see the next one in Line 235 which should be Figure 4, but is referenced as figure 5) in the paper so you need to check and fix these too. In addition to this, the maps in figure 3 need a scale bar.

**Line 272 and 448, 525**. You say here moraines, do you mean ribbed moraine? I think you should call them ribbed moraine other than moraine as they are different.

**Line 460.** Full stop needed after 'poorly captured'

---

## Author Comment (AC1)

Response to reviewers - A machine learning approach to the geomorphometric detection of ribbed moraines in Norway

**Reviewer 2 – Anonymous –** Thank you. We are extremely happy with your acceptance of this paper in an "as is" format. We will of course respond to the comments posed by reviewer 1.

**Reviewer 1 – Paul Dunlop** – Thank you for providing constructive and positive feedback on the manuscript, we greatly appreciate the guidance. Detailed responses to changes will be addressed below, and the manuscript will be adjusted accordingly. In taking account of the edits suggested by Reviewer 1, we have also updated the layout in part to account for new areas of white space thanks to the additional figure.

**Specific Comments and Issues:**

**Line 54.** Is this sentence complete? 'In conjunction with this, the development of new and more robust machine learning methods, coupled with the effectiveness of older methods with new, high–resolution data (clustering and segmentation, Gentleman & Carey, 2008) suggest new approaches for small–scale landform detection'

Should it not end in something like 'suggest new approaches for small–scale landform detection IS NOW POSSIBLE'?

**Added "are now possible".**

**Line 70.** The reference to Lundvist in this line is not factually correct and you need to update your references on this see this from opening paragraph of Dunlop and Clark 2006 in QSR:

Ribbed moraine, also known as Rogen moraine, are subglacially formed transverse ridges that cover extensive areas of the beds of the former Laurentide, Fennoscandian and Irish ice sheets (Fig. 1) and likely exist beneath the Antarctic ice sheet. Originally these landforms were thought to be created at ice margins (Frödin, 1954; Cowan, 1968) hence the use of the word moraine; but it is now widely recognised that they formed subglacially (Hoppe, 1952; Lundqvist, 1969; Aylsworth and Shilts, 1989; Bouchard, 1989; Hättestrand and Kleman, 1999).

I suggest you update this text and include these references from this 2006 paper.

We have updated the text as follows: "Ribbed moraines are subglacial ridges transverse to the glacial flow direction (Dunlop & Clark, 2006), and were documented across Canada and Fennoscandia in the mid-1900s (Hoppe, 1952; Frödin, 1954; Cowan, 1968, Lundqvist, 1969)". The reference list is updated accordingly.

Line 71. This line needs to be rephrased....you state that '*They also are known to form near morphologically similar landforms such as drumlins and hummocks*' They as in ribbed moraine don't not form anything, they are the geomorphological signature of an ice sheet process so it is the formational process that forms similar landforms and drumlins and hummock, so rephrase so it is clearer. Also, you should reference that some authors argue that this suggests that the similarities and transitions from ribs to drumlins suggests there is a bedform continuum and you should reference this. I think it was Aario who first argued

this, see this paper:

RISTO AARIO (1977) Classification and terminology of morainic landforms in Finland, Boreas

https://doi.org/10.1111/j.1502-3885.1977.tb00338.x

But there is evidence in Dunlop and Clark (2006) too for their being a bedform continuum from ribs to drumlins

Rephrased to "They also are known to be present near morphologically similar landforms such as drumlins and hummocks, suggesting related formational processes. This spatial relationship raises the prospect of a "bedform continuum" present beneath ice sheets (Aario, 1977; Lundqvist, 1989; 1997; Dunlop & Clark, 2006; Ely et al., 2016; Möller & Dowling, 2018). Further, this spatial relationship allows us to determine..."

**The reference list has been updated accordingly.**

Line 72 states 'allowing us to test whether our approach can differentiate between different

landforms of similar spatial extent' what does this mean, I think you need to clarify this because you should explain how does your mapping approach test this?

**Okay, clarified as follows:**

Further, this spatial relationship allows us to determine whether our approach can differentiate between different landforms with similar morphologies and spatial extents – this is possible through iteration with ribbed-moraine specific parameters across an area containing "ribbed moraine" and "non-ribbed moraine" landforms.

Line 79. Change the term small-scale landforms to ribbed moraine, as this is what you are mapping and you clarify this by the time the reader gets to here. I suggest doing this throughout.

We agree with this in part – from section 1.3 we will change "small-scale landforms" to "ribbed moraine". In the earlier sections we are aiming to provide more of an overview.

Line 95. I think you could include a few more references here and other examples of how Random Forest has been used in glacial mapping, for example the 2023 paper by Ali et al, in Journal of Glaciology where they mapped glacier change over large parts of the Arctic and high latitudes

We have added the following section and updated the reference list as required: "Random Forest has additionally been proven as a valuable method for glacial and geomorphological work, for example mapping glacier change at high latitudes (Ali et al., 2023) and more directly in geomorphological mapping of glacial and hillslope process produced landforms in Switzerland (Giaccone et al., 2022)."

**Line 99.** Fig 1. I think it would be best to include at least one close up view of what these ribbed moraine look like, you cannot see any, only the mapping outlines. A new figure showing clear unmapped examples is needed in my opinion to show them clearly in the data. You can consider a new composite figure, with the unmapped ribbed moraine, next to the same ones mapped but in a more close up view, this would be a separate and new figure to your current figure one.

This figure has been added as a new figure 1. All figure captions have been fixed accordingly.

Line 100. Change the future tense from 'We will then compare' to 'We then compare'

Resolved.

Line 161. You need to identify what the relevant parameters and morphometrics, what are they?

We have added a reference to figure 3 in this sentence to clear up any uncertainty.

Line 165. In section 1.2 of the paper you discuss a lot about the lack of progression in this field because the high resolution DEM data is not available, then you have the amazing countrywide 1m DEM that overcomes the issue for your study, but it is not clear why you are resampling Norway's 1 m DEM to 10 m making it more course. What is the point of doing this, you need to justify this in your methods as this must impact on the results in some way.

We have added the following:

We resample data for three purposes (a) because ribbed moraine occur at the scale of tens of metres, sub-10 m data is unnecessarily detailed (b) because without resampling the 1 m data captures too much detail in the ground surface, leading to unnecessarily detailed morphometrics, and (c) to limit processing time, as we aim to produce a time-efficient methodology.

**Line 225**. Your figure numbering is wrong here, this should be Figure 3 not 4. This has had a knock on affect on the numbering of figures after this (see the next one in Line 235 which should be Figure 4, but is referenced as figure 5) in the paper so you need to check and fix these too. In addition to this, the maps in figure 3 need a scale bar.

We have reorganized this, and we have added a scale bar on each figure.

Line 272 and 448, 525. You say here moraines, do you mean ribbed moraine? I think you should call them ribbed moraine other than moraine as they are different. Resolved.

**Line 460.** Full stop needed after 'poorly captured' **Resolved**.